# HUBERT UNTANGLES BERT
# TO IMPROVE TRANSFER ACROSS NLP TASKS

## ABSTRACT

We introduce HUBERT[1] which combines the structured-representational power of Tensor-Product Representations (TPRs) and BERT, a pre-trained bidirectional Transformer language model. We show that there is shared structure between different NLP datasets that HUBERT, but not BERT, is able to learn and leverage. We validate the effectiveness of our model on the GLUE benchmark and HANS dataset. Our experiment results show that untangling data-specific semantics from general language structure is key for better transfer among NLP tasks.[2]

## 1 INTRODUCTION

Built on the Transformer architecture (Vaswani et al., 2017), the BERT model (Devlin et al., 2018) has demonstrated great power for providing general-purpose vector embeddings of natural language: its representations have served as the basis of many successful deep Natural Language Processing (NLP) models on a variety of tasks (e.g., Liu et al., 2019a;b; Zhang et al., 2019). Recent studies (Coenen et al., 2019; Hewitt & Manning, 2019; Lin et al., 2019; Tenney et al., 2019) have shown that BERT representations carry considerable information about grammatical structure, which, by design, is a deep and general encapsulation of linguistic information. Symbolic computation over structured symbolic representations such as parse trees has long been used to formalize linguistic knowledge. To strengthen the generality of BERT's representations, we propose to import into its architecture this type of computation.

Symbolic linguistic representations support the important distinction between *content* and *form* information. The form consists of a structure devoid of content, such as an unlabeled tree, a collection of nodes defined by their structural positions or **roles** (Newell, 1980), such as *root*, *left-child-of-root*, *right-child-of-left-child-of root*, etc. In a particular linguistic expression such as *"Kim referred to herself during the speech"*, these purely-structural roles are filled with particular content-bearing symbols, including terminal words like `Kim` and non-terminal categories like `NounPhrase`. These role **fillers** have their own identities, which are preserved as they move from role to role across expressions: `Kim` retains its referent and its semantic properties whether it fills the subject or the object role in a sentence. Structural roles too maintain their distinguishing properties as their fillers change: the *root* role dominates the *left-child-of-root* role regardless of how these roles are filled.

Thus it is natural to ask whether BERT's representations can be usefully factored into content × form, i.e., filler × role, dimensions. To answer this question, we recast it as: can BERT's representations be usefully unpacked into **Tensor-Product Representations (TPRs)**? A TPR is a collection of constituents, each of which is the binding of a filler to a structural role. Specifically, we let BERT's final-layer vector-encoding of each token of an input string be factored explicitly into a filler bound to a role: both the filler and the role are embedded in a continuous vector space, and they are bound together according to the principles defining TPRs: with the tensor product. This factorization effectively untangles the fillers from their roles, these two dimensions having been fully entangled in the BERT encoding itself. We then see whether disentangling BERT representations into TPRs facilitates their general use in a range of NLP tasks.

Concretely, as illustrated in Figure 1, we create HUBERT by adding a TPR layer on top of BERT; this layer takes the final-layer BERT embedding of each input token and transforms it into the tensor

---

[1] HUBERT is a recursive acronym for *HUBERT Untangles BERT.*

[2] Our code and models will be made available after publication.

product of a filler embedding-vector and a role embedding-vector. The model learns to separate fillers from roles in an unsupervised fashion, trained end-to-end to perform an NLP task.

If the BERT representations truly are general-purpose for NLP, the TPR re-coding should reflect this generality. In particular, the formal, grammatical knowledge we expect to be carried by the roles should be generally useful across a wide range of downstream tasks. We thus examine transfer learning, asking whether the roles learned in the service of one NLP task can facilitate learning when carried over to another task.

In brief, overall we find in our experiments on the NLP benchmarks of GLUE (Wang et al., 2018) and HANS (McCoy et al., 2019) that HUBERT's recasting of BERT encodings as TPRs does indeed lead to effective knowledge transfer across NLP tasks, while the bare BERT encodings do not. Specifically, after pre-training on the MNLI dataset in GLUE, we observe positive gains ranging from 0.60% to 12.28% when subsequently fine-tuning on QNLI, QQP, RTE, SST, and SNLI tasks. This is due to transferring TPR knowledge—in particular the learned roles—relative to transferring just BERT parameters which have gains ranging from minus 0.33% to positive 2.53%.

Additionally, on average, we gain 5.7% improvement on the demanding non-entailment class of the HANS challenge dataset. Thus TPR's disentangling of fillers from roles, motivated by the nature of symbolic representations, does yield more general deep linguistic representations as measured by cross-task transfer.

The paper is structured as follows. First we discuss the prior work on TPRs in deep learning and its previous applications in Section 2. We then introduce the model design in Section 3 and present our experimental results in Section 4. We conclude in Section 5.

## 2 RELATED WORK

Building on the successes of symbolic AI and linguistics since the mid-1950s, there has been a long line of work exploiting symbolic and discrete structures in neural networks since the 1990s. Along with Holographic Reduced Representations (Plate, 1995) and Vector-Symbolic Architectures (Levy & Gayler, 2008), Tensor Product Representations (TPRs) provide the capacity to represent the discrete linguistic structure in a continuous, distributed manner, where grammatical form and semantic content can be disentangled (Smolensky, 1990; Smolensky & Legendre, 2006). In Lee et al. (2016), TPR-like representations were used to solve the bAbI tasks (Weston et al., 2016), achieving close to 100% accuracy in all but one of these tasks. Schlag & Schmidhuber (2018) also achieved success on the bAbI tasks, using third-order TPRs to encode and process knowledge-graph triples. In Palangi et al. (2018), a new structured recurrent unit (TPRN) was proposed to learn grammatically-interpretable representations using weak supervision from (context, question, answer) triplets in the SQuAD dataset (Rajpurkar et al., 2016). In Huang et al. (2018), unbinding operations of TPRs were used to perform image captioning. None of this previous work, however, examined the generality of learned linguistic knowledge through transfer learning.

Transfer learning for transformer-based models has been studied recently: Keskar et al. (2019) and Wang et al. (2019) report improvements in accuracy over BERT after training on an intermediate task from GLUE; an approach which has come to be known as Supplementary Training on Intermediate Labeled data Tasks (STILTs). However, as shown in more recent work (Phang et al., 2018), the results do not follow a consistent pattern when using different corpora for fine-tuning BERT, and degraded downstream transfer is often observed. Even for data-rich tasks like QNLI, regardless of the intermediate task and multi-tasking strategy, the baseline results do not improve. This calls for new model architectures with better knowledge transfer capability.

## 3 MODEL DESCRIPTION

Applying the TPR scheme to encode the individual words (or sub-word tokens) fed to BERT, a word is represented as the tensor product of a vector embedding its content—its filler (or symbol)[3] aspect—and a vector embedding the structural role it plays in the sentence. Given the results of

---

[3]We use filler or symbol to indicate the content symbolic representation throughout the paper.

Palangi et al. (2018), we expect the symbol to capture the semantic contribution of the word while the structural role captures its grammatical role:

$$\boldsymbol{x}^{(t)} = \boldsymbol{s}^{(t)} \otimes \boldsymbol{r}^{(t)} \tag{1}$$

Assuming we have $n_S$ symbols with dimension $d_S$ and $n_R$ roles with dimension $d_R$, $\boldsymbol{x}^{(t)} \in \mathbb{R}^{d_S \times d_R}$ is the tensor representation for token $t$, $\boldsymbol{s}^{(t)} \in \mathbb{R}^{d_S}$ is the (presumably semantic) symbol representation and $\boldsymbol{r}^{(t)} \in \mathbb{R}^{d_R}$ is the (presumably grammatical) role representation for token $t$. $\boldsymbol{s}^{(t)}$ may be either the embedding of one symbol or a linear combination of different symbols using a softmax symbol selector, and similarly for $\boldsymbol{r}^{(t)}$. In other words, Eq. 1 can also be represented as $\boldsymbol{x}^{(t)} = \boldsymbol{S}\boldsymbol{B}^{(t)}\boldsymbol{R}^\top$ where $\boldsymbol{S} \in \mathbb{R}^{d_S \times n_S}$ and $\boldsymbol{R} \in \mathbb{R}^{d_R \times n_R}$ are matrices the columns of which contain the global symbol and role embeddings, common for all tokens, and either learned from scratch or initialized by transferring from other tasks, as explained in Section 4. $\boldsymbol{B}^{(t)} \in \mathbb{R}^{n_S \times n_R}$ is the binding matrix which selects specific roles and symbols (embeddings) from $\boldsymbol{R}$ and $\boldsymbol{S}$ and binds them together. We assume that for a single-word representation, the binding matrix $\boldsymbol{B}^{(t)}$ is rank 1, so we can decompose it into two separate vectors, one soft-selecting a symbol and the other a role, and rewrite equation (1) as $\boldsymbol{x}^{(t)} = \boldsymbol{S}(\boldsymbol{a}_S^{(t)}\boldsymbol{a}_R^{(t)\top})\boldsymbol{R}^\top$ where $\boldsymbol{a}_R^{(t)} \in \mathbb{R}^{n_R}$ and $\boldsymbol{a}_S^{(t)} \in \mathbb{R}^{n_S}$ can respectively be interpreted as attention weights over different roles (columns of $\boldsymbol{R}$) and symbols (columns of $\boldsymbol{S}$). For each input token $\boldsymbol{x}^{(t)}$, we get its contextual representations of grammatical role ($\boldsymbol{a}_R^{(t)}$) and semantic symbol ($\boldsymbol{a}_S^{(t)}$) by fusing the contextual information from the role and symbol representations of its surrounding tokens.

We explore two options for mapping the input token from the current time-step, and the tensor representation from the previous time-step, to $\boldsymbol{a}_R^{(t)}$ and $\boldsymbol{a}_S^{(t)}$: a Long Short-Term Memory (LSTM) architecture (Hochreiter & Schmidhuber, 1997) and a one-layer Transformer. All the models share the general architecture depicted in Figure 1 except for BERT and BERT-LSTM, where the TPR layer is absent. Our conclusion based on initial experiments was that the Transformer layer results in better integration and homogeneous combination with the other Transformer layers in BERT, as will be described shortly.

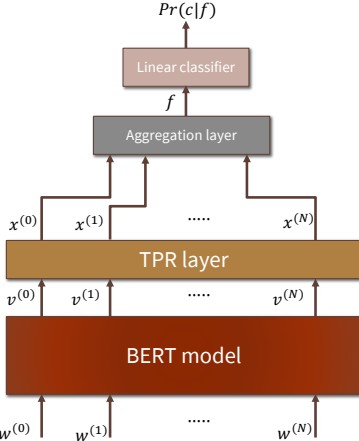

Figure 1: General architecture for all models: HUBERT models have a TPR layer; BERT and BERT-LSTM don't. BERT and TPR layers can be shared between tasks but the classifier is task-specific.

The TPR layer with LSTM architecture works as follows (see also Figure 2, discussed in Sec. 4.2). We calculate the hidden states ($\boldsymbol{h}_S^{(t)}, \boldsymbol{h}_R^{(t)}$) and cell states ($\boldsymbol{c}_S^{(t)}, \boldsymbol{c}_R^{(t)}$) and for each time-step according to the following equations:

$$\boldsymbol{h}_S^{(t)}, \boldsymbol{c}_S^{(t)} = \mathrm{LSTM}_S(\boldsymbol{v}_t, (\mathrm{vec}(\boldsymbol{x}^{(t)}), \boldsymbol{c}_S^{(t-1)})); \quad \boldsymbol{h}_R^{(t)}, \boldsymbol{c}_R^{(t)} = \mathrm{LSTM}_R(\boldsymbol{v}_t, (\mathrm{vec}(\boldsymbol{x}^{(t)}), \boldsymbol{c}_R^{(t-1)})) \tag{2}$$

where $\boldsymbol{v}_t$ is the final-layer BERT embedding of the $t$-th word, and $\mathrm{vec}(.)$ flattens the input tensor into a vector. Each LSTM's input cell state is the previous LSTM's cell output state. Each LSTM's input hidden state, however, is calculated by binding the previous cell's role and symbol vectors.

In the TPR layer with Transformer architecture, we calculate the output representations $(\boldsymbol{h}_{\mathrm{S}}^{(t)}, \boldsymbol{h}_{\mathrm{R}}^{(t)})$ using a Transformer Encoder layer:

$$\boldsymbol{h}_{\mathrm{S}}^{(t)} = \mathrm{Transformer}_S(\boldsymbol{v}_t); \quad \boldsymbol{h}_{\mathrm{R}}^{(t)} = \mathrm{Transformer}_R(\boldsymbol{v}_t) \tag{3}$$

Each Transformer layer consists of a multi-head attention layer, followed by a residual block (with dropout), a layer normalization block, a feed-forward layer, another residual block (with dropout), and a final layer normalization block. (See Figure 3, discussed in Sec. 4.2, and the original Transformer paper, Vaswani et al. (2017), for more details.)

Given that each word is usually assigned to a few grammatical roles and semantic concepts (ideally one), an inductive bias is enforced using a softmax temperature $(T)$ to make $\boldsymbol{a}_{\mathrm{R}}^{(t)}$ and $\boldsymbol{a}_{\mathrm{S}}^{(t)}$ sparse. Note that in the limit of very low temperatures, we will end up with one-hot vectors which pick only one filler and one role.[4]

$$\boldsymbol{a}_{\mathrm{S}}^{(t)} = \mathrm{softmax}(\boldsymbol{W}_{\mathrm{S}}\boldsymbol{h}_{\mathrm{S}}^{(t)}/T); \quad \boldsymbol{a}_{\mathrm{R}}^{(t)} = \mathrm{softmax}(\boldsymbol{W}_{\mathrm{R}}\boldsymbol{h}_{\mathrm{R}}^{(t)}/T) \tag{4}$$

Here $\boldsymbol{W}_{\mathrm{S}}$ and $\boldsymbol{W}_{\mathrm{R}}$ are linear-layer weights. For the final output of the transformer model, we explored different aggregation strategies to construct the final sentence embedding:

$$P(c|f) = \mathrm{softmax}(\boldsymbol{W}_f \mathrm{Agg}(\boldsymbol{x}^{(0)}, \boldsymbol{x}^{(1)}, ..., \boldsymbol{x}^{(N)})) \tag{5}$$

where $P(c|f)$ is a probability distribution over class labels, $f$ is the final sentence representation, $\boldsymbol{W}_f$ is the classifier weight matrix, and $N$ is the maximum sequence length. $\mathrm{Agg}(.)$ defines the merging strategy. We experimented with different aggregation strategies: max-pooling, mean-pooling, masking all but the input-initial [CLS] token, and concatenating all tokens and projecting down using a linear layer. In Devlin et al. (2018), the final representation for the [CLS] token is used as the sentence representation. However, during our experiments, we observed better results when concatenating the final embeddings for all tokens and then projecting down to a smaller dimension, as this exposes the classifier to the full span of token information.

The formal symmetry between symbols and roles evident in Eq. 1 is broken in two ways.

First, we choose hyper-parameters so that the number of symbols is greater than the number of roles. Thus each role is on average used more often than each symbol, encouraging more general information (such as structural position) to be encoded in the roles, and more specific information (such as word semantics) to be encoded in the symbols. (This effect was evident in the analogous TPR learning model of Palangi et al. (2018).)

Second, to enable the symbol that fills any given role to be exactly recoverable from a TPR in which it appears along with other symbols, the role vectors should be linearly independent: this expresses the intuition that distinct structural roles carry independent information. Fillers, however, are not expected to be independent in this sense, since many fillers may have similar meanings and be quasi-interchangeable. So for the role matrix $\boldsymbol{R}$, but not the filler matrix $\boldsymbol{S}$, we add a regularization term to the training loss which encourages the $\boldsymbol{R}$ matrix to be orthogonal:

$$\mathcal{L} = -\sum_c \mathbf{1}[\mathrm{c} = \mathrm{c}^*] \log P(c|f) + \lambda(||\boldsymbol{R}\boldsymbol{R}^\top - \boldsymbol{I}_{d_R}||_F^2 + ||\boldsymbol{R}^\top\boldsymbol{R} - \boldsymbol{I}_{n_R}||_F^2) \tag{6}$$

Here $\mathcal{L}$ indicates the loss function, $\boldsymbol{I}_k$ is the identity matrix with k rows and k columns, and $\mathbf{1}[.]$ is the indicator function: it is 1 when the predicted class $c$ matches the correct class $c^*$ label, and 0 otherwise. Following the practice in Bansal et al. (2018) we use double soft orthogonality regularization to handle both over-complete and under-complete matrices $\boldsymbol{R}$.

---

[4]Note that bias parameters are omitted for simplicity of presentation.

## 4 EXPERIMENTS

We performed extensive experiments to answer the following questions:

1. Does adding a TPR layer on top of BERT (as in the previous section) impact its performance positively or negatively? We are specifically interested in MNLI for this experiment because it is large-scale compared to other GLUE tasks and is more robust to model noise (i.e., different randomly-initialized models tend to converge to the same final score on this task). This task is also used as the source task during transfer learning. This experiment is mainly a sanity check to verify that the specific TPR decomposition added does not hurt source-task performance.

2. Does transferring the BERT model's parameters, fine-tuned on one of the GLUE tasks, help the other tasks in the Natural Language Understanding (NLU) benchmarks (Bowman et al., 2015; Wang et al., 2018)? Based on our hypothesis of the advantage of disentangling content from form, the learned symbols and roles should be transferable across natural language tasks. Does transferring role ($\boldsymbol{R}$) and/or symbol ($\boldsymbol{S}$) embeddings (described in the previous section) improve transfer learning on BERT across the GLUE tasks?

3. Is the ability to transfer the TPR layer limited to GLUE tasks? Can it be generalized? To answer this question we evaluated our models on a challenging diagnostic dataset outside of GLUE called HANS (McCoy et al., 2019).

### 4.1 DATASET

We conduct three major experiments to answer the above questions: a comparison of architectures on the MNLI dataset, a study of transfer learning between GLUE tasks (Wang et al., 2018), and finally model diagnosis using HANS (McCoy et al., 2019); these are discussed in Sections 4.2, 4.3, and 4.4, respectively.

Section A.1 provides a more detailed analysis of the dataset. Table 4 (see Appendix) shows the dataset statistics and details for GLUE, SNLI, and HANS.

### 4.2 ARCHITECTURE COMPARISON ON MNLI

Our experiments are done with four different model architectures. The general architecture of the models is depicted in Figure 1. The TPR layer is absent in BERT and BERT-LSTM. In the figure, the BERT model indicates the pre-trained off-the-shelf BERT base model which has 12 Transformer encoder layers. The aggregation layer computes the final sentence representation (see Eq. 5). The linear classifier is task-specific and is not shared between tasks during transfer learning.

**BERT**: This is our baseline model which consists of BERT, an aggregation layer on top, and a final linear classifier.

**BERT-LSTM**: We augment the BERT model by adding a unidirectional LSTM Recurrent layer (Hochreiter & Schmidhuber, 1997; Sutskever et al., 2014) on top. The inputs to the LSTM are token representations encoded by BERT. We then take the final hidden state of the LSTM and feed it into a classifier to get the final predictions. Since this model has an additional layer augmented on top of BERT, it can serve as a baseline for TPR models introduced below.

**HUBERT (LSTM)**: We use two separate LSTM networks to compute symbol and role representation for each token. Figure 2 shows how the final token embedding ($\boldsymbol{x}^{(t)}$) is constructed at each time step: this plays the role of the LSTM hidden state $\boldsymbol{h}^{(t)}$. (In the figures, '⊛' denotes matrix-vector multiplication.) The results (Table 1) show that this decomposition improves the accuracy on MNLI compared to both the BERT and BERT-LSTM models. Training recurrent models is usually difficult, due to exploding or vanishing gradients, and has been studied for many years (Le et al., 2015; Vorontsov et al., 2017). With the introduction of the gating mechanism in LSTM and GRU cells, this problem was alleviated. In our model, we have a tensor-product operation which binds role and symbol vectors. We observed that during training the values comprising these vectors can reach very small numbers ($< 10^{-4}$), and after binding, the final embedding vectors have values roughly in the order of $10^{-8}$. This makes it difficult for the classifier to distinguish between similar but different sentences. Additionally, backpropagation is not effective since the gradients are too small.

We avoided this problem by linearly scaling all values by a large number ($\sim$ 1K) and making that scaling value trainable so that the model can adjust it for better performance.

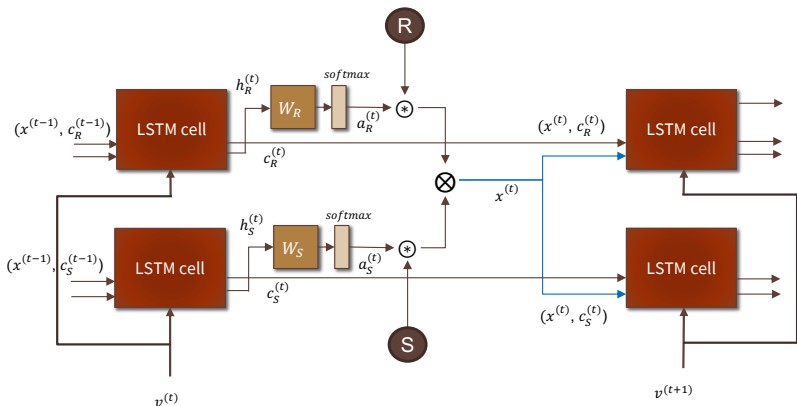

Figure 2: TPR layer architecture for HUBERT (LSTM). $R$ and $S$ are global Role and Symbol embedding matrices which are learned and re-used at each time-step.

**HUBERT (Transformer)**: In this model, instead of using a recurrent layer, we deploy the power of Transformers (Vaswani et al., 2017) to encode roles and symbols (see Figure 3). This lets us attend to all the tokens when calculating $\boldsymbol{a}_R^{(k)}$ and $\boldsymbol{a}_S^{(k)}$ and thus better capture long-distance dependencies. It also speeds up training as all embeddings are computed in parallel for each sentence. Furthermore, it naturally solves the vanishing and exploding gradients problem, by taking advantage of residual blocks (He et al., 2015) to facilitate backpropagation and Layer Normalization (Lei Ba et al., 2016) to prohibit value shifts. It also integrates well with the rest of the BERT model and presents a more homogeneous architecture.[5]

We first do an architecture comparison study on the four models, each built on BERT (base model). We fine-tune BERT on the MNLI task, which we will then use as our primary source training task for testing transfer learning. We report the final accuracy on the MNLI development set.

Table 1 summarizes the results. Both HUBERT models are able to maintain the same performance as our baseline (BERT). This confirms that adding TPR heads will not degrade the model's accuracy and can even improve it (in our case when evaluated on MNLI matched development set). Although HUBERT (Transformer) and HUBERT (LSTM) have roughly the same accuracy, we choose HUBERT (Transformer) to perform our transfer learning experiments, since it eliminates the limitations of HUBERT (LSTM) (as discussed above) and reduces the training and inference time significantly ($>$ 4X).

| Model | BERT | BERT-LSTM | HUBERT (LSTM) | HUBERT (Transformer) |
|---|---|---|---|---|
| Accuracy (%) | 84.15 | 84.17 | 84.26 | 84.30 |

Table 1: MNLI (matched) dev set accuracy for different models.

## 4.3 TRANSFER LEARNING

We compare the transfer-learning performance of HUBERT (Transformer) against BERT. We follow the same training procedure for each model and compare the final development set accuracy on the target corpus. The training procedure is as follows: For Baseline, we train three instances of each model on the target corpus and then select the one with the highest accuracy on target dev set (We vary the random seed and the order in which the training data is sampled for each instance.) These results are reported for each model in the *Baseline Acc.* column in Table 2. For Fine-tuned, in a separate experiment, we first fine-tune one instance of each model on the source corpus and use these

---

[5]The results reported here correspond to an implementation using an additional Transformer encoder layer on top of the TPR layer; we scale the input values to this layer only. Future versions of the model will omit this layer.

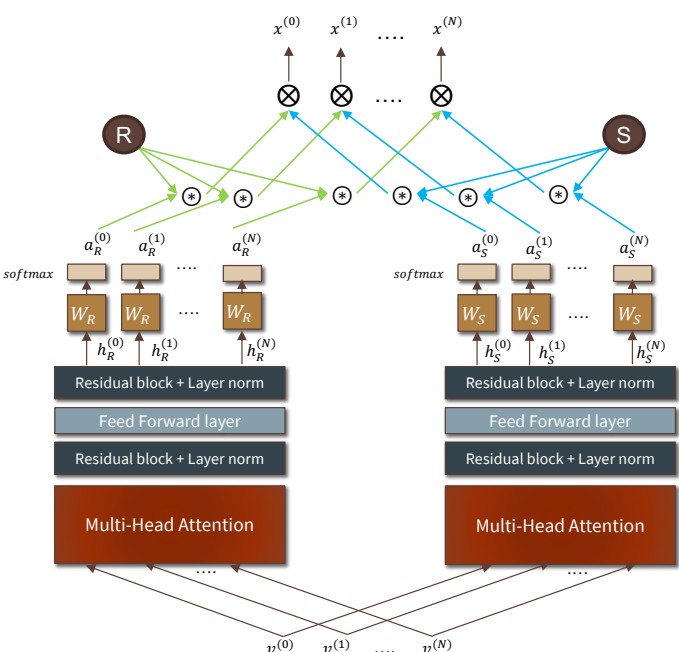

Figure 3: TPR layer architecture for HUBERT (Transformer). $R$ and $S$ are global Role and Symbol embeddings which are learned and shared for all token positions.

updated parameters to initialize a second instance of the same model. The initialized model will then be trained and tested on the target corpus. In this setting, we have three subsets of parameters to choose from when transferring values from the source model to the target model: BERT parameters, the Role embeddings $\boldsymbol{R}$, and the Filler embeddings $\boldsymbol{F}$. Each of these subsets can independently be transferred or not, leading to a total of 7 combinations excluding the option in which none of them are transferred. We chose the model which has the highest absolute accuracy on the target dev dataset. These results are reported for each model under *Fine-tuned Acc*. Note that the transferred parameters are not frozen, but updated during training on the target corpus.

**MNLI as source**: Table 2 summarizes the results for these transfer learning experiments when the source task is MNLI. *Gain* shows the difference between Fine-tuned model's accuracy and Baseline's accuracy. For HUBERT (Transformer), we observe substantial gain across all 5 target corpora after transfer. However, for BERT we have a drop for QNLI, QQP, and SST.

These observations confirm our hypothesis that recasting the BERT encodings as TPRs leads to better generalization across down-stream NLP tasks.

Almost all tasks benefit from transferring roles except for QNLI. This may be due to the structure of this dataset, as it is a modified version of a question-answering dataset (Rajpurkar et al., 2016) and has been re-designed to be an NLI task. Transferring the filler embeddings helps with only QNLI and RTE. Transferring BERT parameters in conjunction with fillers or roles surprisingly boosts accuracy for QNLI and SST, where we had negative gains for the BERT model, suggesting that TPR decomposition can also improve BERT's parameter transfer.

**QQP as source**: The patterns here are quite different as the source task is now a paraphrase task (instead of inference) and TPR needs to encode a new structure. Again transferring roles gives positive results except for RTE. Filler vectors learned from QQP are more transferable compared to MNLI and gives a boost to all tasks except for SNLI. Surprisingly, transferring BERT parameters is hurting the results now even when TPR is present. However, for cases in which we also transferred BERT parameters (not shown), the Gains were still higher than for BERT, confirming the results obtained when MNLI was the source task.[6]

---

[6]Baseline results slightly differ from Table 2 due to using a different scaling value for this each source task.

| Model | Target Corpus | Transfer BERT | Transfer Filler | Transfer Role | Baseline Acc. (%) | Fine-tuned Acc. (%) | Gain (%) |
|---|---|---|---|---|---|---|---|
| BERT | QNLI | True | – | – | 91.60 | 91.27 | − 0.33 |
| BERT | QQP | True | – | – | 91.45 | 91.12 | − 0.33 |
| BERT | RTE | True | – | – | 71.12 | 73.65 | + 2.53 |
| BERT | SNLI | True | – | – | 90.45 | 90.69 | + 0.24 |
| BERT | SST | True | – | – | 93.23 | 92.78 | − 0.45 |
| HUBERT (Transformer) | QNLI | True | True | False | 90.56 | 91.16 | **+ 0.60** |
| HUBERT (Transformer) | QQP | False | False | True | 90.81 | 91.42 | **+ 0.61** |
| HUBERT (Transformer) | RTE | True | True | True | 61.73 | 74.01 | **+ 12.28** |
| HUBERT (Transformer) | SNLI | True | False | True | 90.66 | 91.36 | **+ 0.70** |
| HUBERT (Transformer) | SST | True | False | True | 91.28 | 92.43 | **+ 1.15** |

Table 2: Transfer learning results for GLUE tasks. The source corpus is MNLI. Baseline accuracy is when Transfer BERT, Filler, and Role are all False, equivalent to no transfer. Fine-tuned accuracy is the best accuracy among all possible transfer options.

We also verified that our TPR layer is not hurting the performance by comparing the *test* set results for HUBERT (Transformer) and BERT. The results are obtained by submitting models to the GLUE evaluation server. The results are presented in Table 6.

## 4.4 MODEL DIAGNOSIS

We also evaluated HUBERT (Transformer) on a probing dataset outside of GLUE called HANS (McCoy et al., 2019) Results are presented in Table 3. HANS is a diagnosis dataset that probes various syntactic heuristics which many of the state-of-the-art models turn out to exploit, and thus they perform poorly on cases that don't follow those heuristics. There are three heuristics measured in HANS which are as follows: *Lexical overlap* where a premise entails any hypothesis built from a subset of words in the premise, *Subsequence* where a premise entails any contiguous subsequences of it, and *Constituent* where a premise entails all complete subtrees in its parse tree. Our results indicate that TPR models are less prone to adopt these heuristics, resulting in versatile models with better domain adaptation. Following McCoy et al. (2019), we combined the predictions of *neutral* and *contradictory* into a *non-entailment* class, since HANS uses two classes instead of three. Note that no subset of the HANS data is used for training.[7]

We observed that our HUBERT (Transformer) model trained on MNLI did not diminish BERT's near-perfect performance on correctly-entailed cases (which follow the heuristics). In fact, it increased the accuracy of Lexical and Subsequence heuristics. On the problematic Non-Entailment cases, however, BERT outperforms HUBERT (Transformer). Since HUBERT has more parameters than BERT it can better fit the training data. Thus, we suspect that HUBERT attends more to the heuristics that MNLI has in its design, and gets a lower score on sentences that don't follow those heuristics. But to examine the knowledge-transfer power of TPR, we additionally fine-tuned each model on SNLI and tested again on HANS. (For HUBERT (Transformer), we only transfer roles and fillers). On Non-entailment cases, for the HUBERT model, the Lexical accuracy improved drastically: by 61.62% (6,162 examples). Performance on cases violating the Subsequence heuristic improved by 1.44% (144 examples) and performance on those violating the Constituent heuristic improved by 5.4% (540 examples). These improvements on Non-entailment case came at the cost of small drops in Entailment accuracy. This pattern of transfer is in stark contrast with the BERT results. Although the results on Entailment cases are improved, the accuracies for Subsequence and Constituent Non-Entailment cases drop significantly, showing that BERT is failing to integrate new knowledge gained from SNLI with previously learned information from MNLI. This shows that here, HUBERT (Transformer) can leverage information from a new source of data efficiently. The huge improvement on the Lexical Non-entailment cases speak to the power of TPRs to generate role-specific word embeddings: the Lexical heuristic amounts essentially to performing inference

---

[7]We observed high variance in the results on HANS for both BERT and HUBERT. For instance, two models that achieve similar scores on the MNLI dev set can have quite different accuracies on HANS. To account for this, we ran our experiments with at least 3 different seeds and reported the best scores for each model.

on a bag-of-words representation, where mere lexical overlap between a premise and a hypothesis yields a prediction of entailment.

| Model | Acc. (%) | Correct: Entailment | | | Correct: Non-Entailment | | |
|---|---|---|---|---|---|---|---|
| | | Lex. (%) | Sub. (%) | Const. (%) | Lex. (%) | Sub. (%) | Const. (%) |
| BERT | 63.59 | 95.32 | 99.32 | 99.44 | 53.40 | 8.86 | 25.20 |
| BERT + | 61.03 ↓ | 98.70 ↑ | 99.96 ↑ | 100.00 ↑ | 55.22 ↑ | 2.92 ↓ | 9.40 ↓ |
| HUBERT (Transformer) | 52.31 | 98.30 | 99.92 | 99.40 | 8.40 | 2.32 | 5.52 |
| HUBERT (Transformer) + | 63.22 ↑ | 95.52 ↓ | 99.76↓ | 99.32 ↓ | 70.02 ↑ | 3.76 ↑ | 10.92 ↑ |

Table 3: HANS results for BERT and HUBERT (Transformer) models. Acc. indicates the average of the results on each sub-task in HANS. Each model is fine-tuned on MNLI. '+' indicates that the model is additionally fine-tuned on the SNLI corpus. ↑ indicates an increase and ↓ indicates a decrease in accuracy after the model is fine-tuned on SNLI.

## 5 CONCLUSION

In this work we showed that BERT cannot effectively transfer its knowledge across NLP tasks, even if the two tasks are fairly closely related. To resolve this problem, we proposed HUBERT: this adds a decomposition layer on top of BERT which disentangles symbols from their roles in BERT's representations. The HUBERT architecture exploits Tensor-Product Representations, in which each word's representation is constructed by binding together two separated properties, the word's (semantic) content and its structural (grammatical) role. In extensive empirical studies, HUBERT showed consistent improvement in knowledge-transfer across various linguistic tasks. HUBERT+ outperformed BERT+ on the challenging HANS diagnosis dataset, which attests to the power of its learned, disentangled structure. The results from this work, along with recent observations reported in Kovaleva et al. (2019); McCoy et al. (2019); Clark et al. (2019); Michel et al. (2019), call for better model designs enabling synergy between linguistic knowledge obtained from different language tasks.

### ACKNOWLEDGMENTS

We would like to thank R. Thomas McCoy from Johns Hopkins University and Alessandro Sordoni from Microsoft Research for sharing and discussing their recent results on HANS, and Xiaodong Liu from Microsoft Research for thoughtful discussions.

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

## A    APPENDIX

### A.1    DATASET DETAILS

In this section, we briefly describe the datasets we use to train and evaluate our model. GLUE is a collection of 9 different NLP tasks that currently serve as a good benchmark for different proposed language models. The tasks can be broadly categorized into single sentence tasks (e.g. CoLA and SST) and paired sentence tasks (e.g. MNLI and QQP). In the former setting, the model makes a binary decision on whether a single input satisfies a certain property or not. For CoLA, the property is grammatical acceptability; for SST, the property is positive sentiment.

The 7 other tasks in GLUE are paired sentence tasks in which the model strives to find a relationship (binary or ternary) between two sentences. QNLI, WNLI, and RTE are inference tasks, in which given a *premise* and a *hypothesis*, the model predicts whether the hypothesis is congruent with the premise (i.e. entailment) or not (i.e. conflict). Although QNLI and WNLI are not originally designed as inference tasks, they have been re-designed to have a similar configuration as other

NLI tasks. This way, a single classifier can be used to judge whether the right answer is in the hypothesis (e.g. for QNLI) or whether a pronoun is replaced with the correct antecedent (e.g. for WNLI). MNLI is an additional NLI task in which three classes are being used instead of two to represent the relation between two sentences. The third class shows neutrality when the model is not confident that the relation is either entailment or contradiction. The last three tasks measure sentence similarity. In MRPC the model decides if two sentences are paraphrases of each other. In QQP, given two questions, the model decides whether they are equivalent and are asking for the same information. All the tasks discussed so far fall under the classification category, where the model produces a probabilistic distribution over the possible class outcomes and the highest value is selected. STS-B, however, is a regression task where the model produces a real number between 1 and 5, indicating the two sentences' semantic similarity. Since our model is designed only for classification tasks, we skip this dataset.

| Corpus | Task | single\pair | # Train | # Dev | # Test | # Labels |
|--------|------|-------------|---------|-------|--------|----------|
| CoLA | Acceptability | single | 8.5K | 1K | 1K | 2 |
| SST | Sentiment | single | 67K | 872 | 1.8K | 2 |
| MRPC | Paraphrase | pair | 3.7K | 408 | 1.7K | 2 |
| QQP | Paraphrase | pair | 364K | 40K | 391K | 2 |
| MNLI | Inference | pair | 393K | 20K | 20K | 3 |
| QNLI | Inference | pair | 108K | 5.7K | 5.7K | 2 |
| RTE | Inference | pair | 2.5K | 276 | 3K | 2 |
| WNLI | Inference | pair | 634 | 71 | 146 | 2 |
| SNLI | Inference | pair | 549K | 9.8K | 9.8K | 3 |
| HANS | Inference | pair | – | – | 30K | 2 |

Table 4: Details of the GLUE (excluding STS-B), SNLI and HANS corpora

We observed a lot of variance in the accuracy ($\pm 5\%$) for models trained on WNLI, MRPC, and CoLA. As mentioned in the GLUE webpage[8], there are some issues with the dataset, which makes many SOTA models perform worse than majority-voting. We found that MRPC results are highly dependent on the initial random seed and order of sentences in the shuffled training data which is mainly caused by the small number of training samples (Table 4). CoLA is the only task in GLUE which examines grammatical correctness rather than sentiment, and thus it makes it harder to benefit from the knowledge learned from other tasks. The train and test set are also constructed in an adversarial way which makes it very challenging. For example, the sentence "*Bill pushed Harry off the sofa for hours.*" is labeled as incorrect in the train split but a very similar sentence "*Bill pushed Harry off the sofa.*" is labeled as correct in the test split. Hence, we only conduct our experiments on the remaining 5 datasets from GLUE.

We also take advantage of an additional NLI dataset called SNLI. It is distributed in the same format as MNLI and recommended by Wang et al. (2018) to be used in conjunction with MNLI during training. However, in our experiments, we treat this dataset as a separate corpus and report our results on it individually.

To further test the capabilities of our model, we evaluate our model on a probing dataset (McCoy et al., 2019). It introduces three different syntactic heuristics and claims that most of SOTA neural NLI models exploit these statistical clues to form their judgments on each example. It shows through extensive experiments that these models obtain very low accuracies for sentences cleverly crafted to defeat the models which exploit these heuristics. Lexical overlap, Subsequence, and Constituent are the three categories examined, each containing 10 sub-categories.

## A.2 TEST RESULTS

Table 5 shows the transfer learning results when the source corpus is QQP. Table 6 shows the test results for BERT and HUBERT (Transformer). The top 5 rows are for MNLI as source corpus and bottom 5 rows are for QQP as source corpus.

---

[8]https://gluebenchmark.com/faq

| Model | Target Corpus | Transfer BERT | Transfer Filler | Transfer Role | Baseline Acc. (%) | Fine-tuned Acc. (%) | Gain (%) |
|---|---|---|---|---|---|---|---|
| BERT | QNLI | True | – | – | 91.60 | 90.96 | − 0.64 |
| BERT | MNLI | True | – | – | 84.15 | 84.41 | + 0.26 |
| BERT | RTE | True | – | – | 71.12 | 62.45 | − 8.67 |
| BERT | SNLI | True | – | – | 90.45 | 90.88 | + 0.43 |
| BERT | SST | True | – | – | 93.23 | 92.09 | − 1.14 |
| HUBERT (Transformer) | QNLI | False | True | True | 88.32 | 90.55 | **+ 2.23** |
| HUBERT (Transformer) | MNLI | False | True | True | 84.30 | 85.24 | **+ 0.94** |
| HUBERT (Transformer) | RTE | False | True | False | 61.73 | 65.70 | **+ 3.97** |
| HUBERT (Transformer) | SNLI | False | False | True | 90.63 | 91.20 | **+ 0.57** |
| HUBERT (Transformer) | SST | True | True | True | 86.12 | 91.06 | **+ 4.94** |

Table 5: Transfer learning results for GLUE tasks. The source corpus is QQP. Baseline accuracy is for when Transfer BERT, Filler, and Role are all False, which is equivalent to no transfer. Fine-tuned accuracy is the best accuracy among all possible transfer options.

| Source Corpus | Target Corpus | HUBERT Transfer BERT | HUBERT Transfer Filler | HUBERT Transfer Role | BERT Acc. (%) | HUBERT Acc. (%) |
|---|---|---|---|---|---|---|
| MNLI | QNLI | True | True | False | **90.50** | **90.50** |
| MNLI | QQP | False | False | True | 89.20 | **89.30** |
| MNLI | RTE | True | True | True | 66.40 | **69.30** |
| MNLI | SNLI | True | False | True | 89.20 | **90.35** |
| MNLI | SST | True | False | True | **93.50** | 92.60 |
| QQP | QNLI | False | True | True | 90.50 | **90.70** |
| QQP | MNLI | False | True | True | 84.60 | **84.70** |
| QQP | RTE | False | True | False | **66.40** | 63.20 |
| QQP | SNLI | False | False | True | 89.20 | **90.36** |
| QQP | SST | True | True | True | **93.50** | 91.00 |

Table 6: Test set results for HUBERT (Transformer) and BERT. BERT accuracy indicates test results on target corpus (without transfer) for *bert-base-uncased* which are directly taken from the GLUE leaderboard. Fine-tuned accuracy are the test results for best performing HUBERT (Transformer) model on target dev set after transfer (see Tables 2 and 5).

### A.3 IMPLEMENTATION DETAILS

Our implementations are in PyTorch and based on the HuggingFace[9] repository which is a library of state-of-the-art NLP models, and BERT's original codebase[10]. In all of our experiments, we used `bert-base-uncased` model which has 12 Transformer Encoder layers with 12 attention heads each and the hidden layer dimension of 768. BERT's word-piece tokenizer was used to preprocess the sentences. We used Adamax (Kingma & Ba, 2014) as our optimizer with a learning rate of $5 \times 10^{-5}$ and used a linear warm-up schedule for 0.1 proportion of training. In all our experiments we used the same value for dimension and number of roles and symbols ($d_S$: 32, $d_R$: 32, $n_S$: 50, $n_R$: 35). These parameters were chosen from the best performing BERT models over MNLI. We used the gradient accumulation method to speed up training (in which we accumulate the gradients for two consecutive batches and then update the parameters in one step). Our models were trained with a batch size of 256 distributed over 4 V100 GPUs. Each model was trained for 10 epochs, both on the source task and the target task (for transfer learning experiments).

We performed hyper-parameter tuning for both BERT and HUBERT models on the MNLI dev set. As for the dimension of roles and symbols we did grid search over these values: [10, 30, 60] We fixed the number of roles to 35 and searched among these values for number of fillers: [50, 100, 150]. We additionally performed some light tuning on learning rate, temperature value, and scaling value. To control for the randomness in our results, we ran our experiments by fine-tuning BERT with 3 different seeds and choosing the best results among them. However, for HUBERT we used the same seed for all 7 experiments and only changed the initial weights of layers in the model. We observed small variance in baseline and fine-tuned accuracy across different runs for BERT model. Specifically when MNLI is the source corpus, the standard deviation for QNLI, QPP, RTE, SNI, and SST as target corpora was 0.2%, 0.4%, 0.5%, 0.4%, 0.4% respectively.

### A.4 INTERPRETATION OF LEARNED ROLES

To gain a better understanding of what $R$ (the global role matrix) is learning we analyze the attention scores ($a_R$) over this matrix generated by the model for each token. The final role vector ($r^{(t)}$) for each token is being calculated by performing a matrix product between the attention vector and Role matrix as discussed in Section 3:

$$r^{(t)} = R a_{\mathrm{R}}^{(t)} \tag{7}$$

$a_R$ is a vector of $nR$ dimension indicating the importance of each Role in constructing the final role vector $r^{(t)}$. First, for each sentence in the dataset we collect the Part Of Speech (POS) tags for each token (or sub-token) coming out of BERT model. We obtain the POS tags from the last sub-tokens of each word. This requires us to take extra care when processing the data and keep a dictionary that maps the index of each token in the original sentence to the indices of all its sub-tokens. Then, we gather the roles each sub-token is being attracted to. To account for the distributed nature of the attention scores, instead of choosing the role with highest value in the attention distribution, we select the top K values, concatenate them, and treat this new tuple as a single role. We then calculate the number of roles assigned to each POS tag. Figure 4 shows this distribution where each color shows a specific role tuple.

The number of all possible POS tags in MNLI is 36 which nicely aligns with the 35 number of roles we chose in our experiments. See this link[11] for a description of each tag. To make the visualizations simpler we merged tags indicating similar grammar roles into one coming up with a total of 21 tags (e.g. combining NN, NNS, NNP, and NNPS into one category: NN).

We can observe some interesting patterns by looking at the frequency of the roles assigned to each tag. For example, we observe that the "yellow" role is attracted to '.' tag which represents punctuation marks such as a full stop, question mark, exclamation mark, etc. On the other hand, the "blue" role is attracted almost exclusively to NN. Additionally the "green" role is mainly present in NN, VB, and JJ which are linguistically the three most important POS categories. However, the "purple"

---

[9] https://github.com/huggingface/pytorch-pretrained-BERT

[10] https://github.com/google-research/bert

[11] Penn Treebank POS

role is more or less used by all different POS categories suggesting low correlation between that role and POS tags.

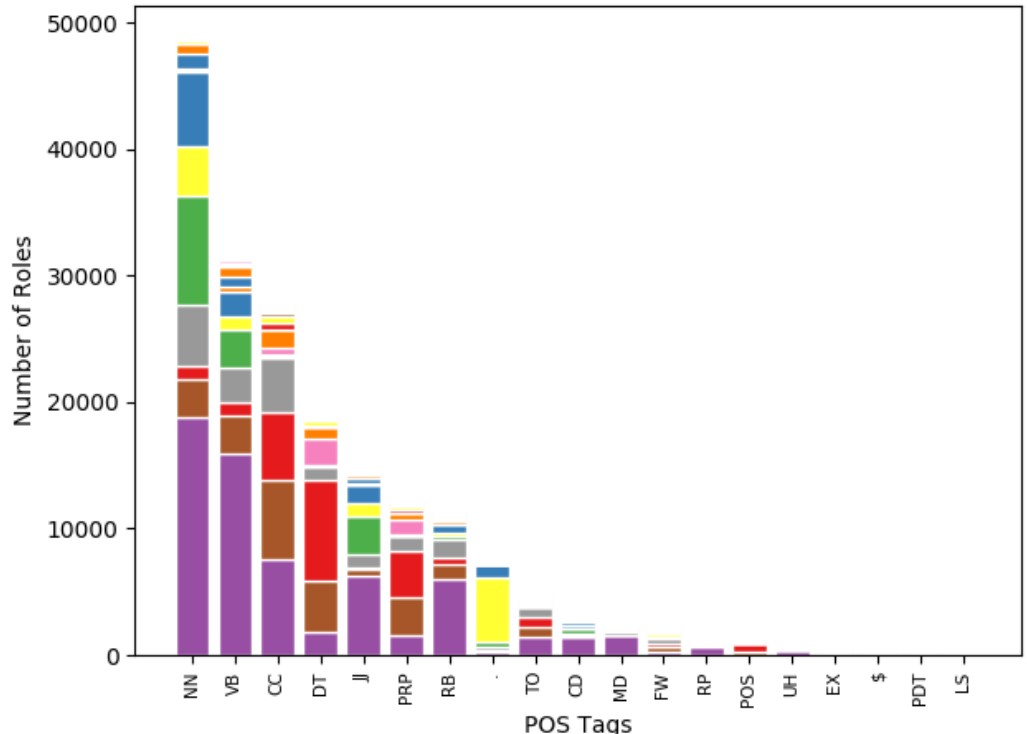

Figure 4: POS tags vs Role frequencies when selecting the two roles with highest value in the attention distribution. Subsequently the tags referring to similar grammar roles are merged into one category to generate better visualization.

