# OpenReview forum: "HUBERT Untangles BERT to Improve Transfer across NLP Tasks"
_ICLR.cc/2020/Conference — Reject_

### Official Review · AnonReviewer3 · 2019-10-18
**Official Blind Review #3**

**Rating:** 1

**Review:**

This paper proposes a fine-tune technique to help BERT models to learn & capture form and content information on textual data (without any form of structural parsing needed). They key addition to the classic BERT model is the introduction of the R and S embeddings. R &S are supposed to learn the information in text that is traditionally represented as the structural positions and the content-bearing  symbols in those positions.

In order to effectively learn R and S embeddings, the authors propose two possible ways to do so: LSTM (Fig 2) and 1-layer Transformer (Fig 3). The main experiments are based on 1-layer transformer HUBERT b/c from a single test in Table 1, the transformer variant appears to be working better than the LSTM variant.

My main concern regarding this paper is two-fold: limited novelty and insignificant performance gain.
The authors did a great job motivating the need for separating role and filler in the intro. However, in neither implementation of HUBERT, I do not see how the structural information (e.g., a parse tree) is directly incorporated into the learning of HUBERT.

Regarding the performance, it seems HUBERT is gaining very little over the BERT baseline. please refer to my specific question below.


Questions:
What are the numeric values for d_S, d_R, n_S, n_R (defined under Section 3 on page 2) in experiment ? I think d_S, d_R are determined at author's discretion (just like the dimensionality of, say, the LSTM hidden layer). But how are n_S and n_R determined?

Page 7, first paragraph: what is Filler embeddings F? F is not defined in either version the proposed HUBERT( Figure 2 or Figure 3). Did the authors mean S?

Table 2. Why do the first 5 rows and the bottom 5 rows have different baseline Acc. ? Shouldn't we always use the best accuracy as baseline for comparison? If we look at the HUBERT Fine-tuned Acc., in many cases, they are actually worse than the best baseline acc. available. (i.e., QNLI , QQP, and SST).

Other comments:
Typo on page one: “[] To strengthen the generality of ….”
Figure 1 is never referred in main text.


**Experience Assessment:**

I have read many papers in this area.

**Review Assessment: Checking Correctness Of Derivations And Theory:**

I carefully checked the derivations and theory.

**Review Assessment: Checking Correctness Of Experiments:**

I carefully checked the experiments.

**Review Assessment: Thoroughness In Paper Reading:**

I read the paper at least twice and used my best judgement in assessing the paper.

---

> ### Author Response · Authors · 2019-11-15
> **Response to AnonReviewer3**
>
> We would like to thank you for the comments and feedback.
>
> In this work, we propose a new model combining the power of deep neural language models such as BERT with symbolic representations such as Tensor-Product Representations. To the best of our knowledge, this is the first work that examines implicit structure learning of transformer-based models on NLP tasks and provides a different way of doing transfer learning among different corpora in GLUE. We also show that this architecture can benefit other out of distribution probing tasks by achieving 2.21% absolute improvement in accuracy on the HANS dataset.
>
> As you pointed out, our main motivation is separating data-specific semantics from general sentence structure by the means of a TPR layer. This is done in an unsupervised way and thus we don’t inject any prior information on what roles or fillers should be learned.
>
> Response to your questions:
>
> 1)
> We have reported the implementation details including d_S, d_R, n_S, n_R values in section A.3 of the paper under the implementation details section.
> We performed hyper-parameter tuning for both BERT and HUBERT models.
>  As for the dimension of roles and symbols, we did grid search over these values: [10, 30, 60]
> We fixed the number of roles to 35 and searched among these values for the number of fillers: [50, 100, 150].
> We chose the final values according to the best performance on MNLI dev set.
> LSTM cell's hidden dimension for LSTM models is set to BERT base model hidden dimension which is 768 and for HUBERT (LSTM) is set to "dS * dR = 32 * 32" to eliminate the need for a projection layer when calculating new hidden states.
>
> 2)
> Thank you for pointing this out. We have used symbol and filler notation interchangeably throughout the paper based on the context. We have added a footnote addressing this.
>
> 3)
> The top 5 rows in Tables 2 and 3 show the performance of BERT model on 5 different target tasks. The baseline accuracies are when the model is initialized randomly and trained and tested on the target task. Consequently, these accuracies are different because they are measured for different tasks. The fine-tuned accuracy corresponds to a model that is fine-tuned on MNLI and then trained and tested on the target task.
> This happens for the bottom 5 rows of each Table showing the result for HUBERT.
> We report the best performing model on the target task dev set, and thus we indeed use the best accuracy as a baseline for comparison.
> Although the baseline results for HUBERT are slightly worse than BERT in Table 2, we show that HUBERT performs much better when initialized with a fine-tuned model (compared to its own baseline), whereas BERT sometimes degrades the performance after knowledge transfer.
>
>
> response to the minor comments:
> - Thank you for spotting the typo. It is now fixed in the revision.
> - Figure 1 is being referred to in the introduction and Section 4.2 (Ablation Study). However, we will move it up to page 3, so that readers can refer to it earlier.

---

### Official Review · AnonReviewer1 · 2019-10-22
**Official Blind Review #1**

**Rating:** 3

**Review:**

This paper proposes a layer on top of BERT which is motivated by a desire to disentangle content (meaning of the tokens) and form (structural roles of the tokens).  Figure 1 shows this clearly. The paper considers two variants of the disentangling layer (TPR), one with LSTMs (figure 2) and the other with attention (figure 3). The aim in both is to obtain a decomposition of the form x(t) = S a_s(v_t) a_r(v_t) R where S and R are shared matrices of parameters and v is the output of BERT.

The model is well motivated and includes clear reasonable design ideas, including choosing hyper-parameters so that the number of symbols (s) is greater than the number of roles (r), and forcing only the roles to be independent (eqn 6).

Minor: I would have preferred that figure 1 appeared earlier in page 3. This would help as the authors forgot to define v in eqn 2. One has to wait for the figure. Having said this, the paper is extremely clear in the notation and does an excellent job at defining dimensions for all the quantities of interest.

I read the paper eagerly and with excitement until I got to the results. First, it wasn't clear to me how well motivated is the idea of fine-tuning on intermediate tasks. I understand the authors are just trying to make a point that BERT does worse than their model in this case and that this is not good for transfer, but still I find this to be artificially constructed.

 The variations in the numbers seem small and possibly attributable to other factors. For this reason, I feel the authors should have continued showing results for the other baselines from the first experiment. I would also have loved to see some visualizations for a, r, A and R in the appendix. Some visualization and anecdotal results might have helped me see that the motivation is backed up by the results. I hope the authors have the time to do this and consider the extra experiments.

**Experience Assessment:**

I have published one or two papers in this area.

**Review Assessment: Checking Correctness Of Derivations And Theory:**

I assessed the sensibility of the derivations and theory.

**Review Assessment: Checking Correctness Of Experiments:**

I assessed the sensibility of the experiments.

**Review Assessment: Thoroughness In Paper Reading:**

I read the paper at least twice and used my best judgement in assessing the paper.

---

> ### Author Response · Authors · 2019-11-15
> **Response to AnonReviewer1**
>
> We would like to thank you for your review. Your comments on the work are much appreciated!
>
> - As correctly pointed out, our work shows improvement in transfer learning across different tasks in GLUE. Please note that fine-tuning BERT model on intermediate tasks and evaluating its transferability is a challenging problem. For example, see the following papers (one of which is from the GLUE authors): https://arxiv.org/abs/1811.01088
> , https://arxiv.org/abs/1812.10860
> They report that BERT (and other transformer-based models) have inconsistent results when transferring knowledge from an intermediate task to the target task, and often impact the down-stream task results negatively. This confirms our findings in this work and supports the importance of transferability among NLP tasks when finetuned on an intermediate task.
>
> - To control for the randomness in the transfer learning results we ran our experiments by fine-tuning BERT with 3 different seeds and choosing the best results among them. However, for HUBERT we used the same seed for all 7 experiments and only changed the initial weights of layers in the model. We added more information regarding experiment settings in the new section we added to the paper, Section A.2 (Implementation details) and discussed the variance of the observed results.
>
> - Although the baseline results for HUBERT are slightly worse than BERT in Tables 2 and 3, it is evident that HUBERT performs much better when initialized with a fine-tuned model (compared to its own baseline), whereas BERT usually degrades the performance after knowledge transfer.
>
> - We added a section in the appendix (A.4) showing the interpretation and visualization of learned roles. Please refer to the updated version of the paper.
>
>
> - response to the minor comment:
>   Thank you for your comment. We have now moved Fig. 1 to page 3 as you suggested.

---

### Official Review · AnonReviewer2 · 2019-10-27
**Official Blind Review #2**

**Rating:** 3

**Review:**

This paper proposes an alternative way of reusing pretrained BERT for downstream tasks rather than the traditional method of fine-tuning the embeddings equivalent to the CLS token.

For each bert embedded token, the proposed method aims at disentangling semantic information of the word from its structural role. Authors provide two ways to provide this disentagling using LSTM or transformer blocks. with several design choices such as: *  a regularization term to encourages the roles matrix to be orthogonal and hence each role carry independent information *  design the roles and symbols matrices so that the number of symbols is greater than the number of roles

In evaluation authors design several experiments to show that:
* Does transferring disentangled role & symbol embeddings improve transfer learning
* the effectiveness of the TPR layer on performance?
* Transfer beyond Glue tasks?

While those experiments provide empirical gains of the design choices, authors don't show enough study to attribute those  empirical gains to the presented design choices:

One large claim in the paper is that empirical gains in the ability of transfer between similar tasks MNLI and GLUE is because of disentangling the semantics from the role representations. We don't know if the TPR layer really manages to do that, this could have been easily verified using for example clustering word senses of the same word.

The empirical gains in transfer learning can be simply attributed to:
- More params it seems adding an LSTM over bert embeddings already does some improvement, I would have loved to see this more exploited but it wasn't. This aligns with some recent findings that BERT is undertrained (Liu et al. 2019) https://arxiv.org/abs/1907.11692
- Variance in the results (authors report only results of one single run not mean and std of several runs).
- More budget given to hyper-parameter search for the models proposed in the paper.  Hyper param budget isn't also reported in the paper.
- other factors, not the ones associated with the claims in the paper: for example what authors claim is an ablation study was comparing several different models together. It would have been more interesting to see for example the effect of making the # symbols = # roles or removing the orthogonality loss from the roles matrix.

Conclusion: The paper introduces large claims and empirical results that correlate with, however the provided experiments are not done with enough control to attribute gains to the design choices provided in the paper.

**Experience Assessment:**

I have published in this field for several years.

**Review Assessment: Checking Correctness Of Derivations And Theory:**

I carefully checked the derivations and theory.

**Review Assessment: Checking Correctness Of Experiments:**

I assessed the sensibility of the experiments.

**Review Assessment: Thoroughness In Paper Reading:**

I read the paper at least twice and used my best judgement in assessing the paper.

---

> ### Author Response · Authors · 2019-11-15
> **Response to AnonReviewer2**
>
> Thanks for your detailed and helpful feedback!
>
> We address each of your comments regarding the empirical gains in transfer learning below:
>
> On more parameters for other models:
>
> In our initial experiments, we performed an ablation study by inserting a TPR or LSTM layer on top of certain layers of BERT (e.g. first 2 layers) and omitting the remaining layers. In those experiments, we observed that LSTM was degrading the performance whereas TPR was improving it. However, this conclusion is not True when all 12 layers of BERT-Base are used. For example, the MNLI dev accuracy of the LSTM model with only 10 layers of BERT was 82.64%, 0.84% lower than the accuracy of just the BERT model with no LSTM heads.
> Therefore, we observed having more parameters does not necessarily result in better accuracies especially when the added layer is not pretrained with the rest of the model.
>
> On variance in the results:
>
> We controlled for the randomness in the results in Table 1 by fine-tuning BERT and HUBERT with 3 different seeds in our experiments and choosing the best results among them. For the cases in which BERT has negative gains after transfer, we observed the same trend, independent of the random seed used. For all other target tasks except SNLI, the mean value for HUBERT gains were always higher than BERT gains. We have added notes regarding the variance of the results in section A.3 (Implementation Details).
>
> On more budget for hyper-parameter search:
>
> We performed hyper-parameter tuning for both BERT and HUBERT models on the MNLI dev set.
> As for the dimension of roles and symbols, we did grid search over these values: [10, 30, 60]
> We fixed the number of roles to 35 and searched among these values for the number of fillers: [50, 100, 150].
> We additionally performed some light tuning on learning rate, temperature value, and scaling value.
> Please refer to section A.3 in the appendix for implementation details.
>
>
> On other contributing factors:
>
> We carried out experiments by changing the ratio of fillers and roles. Making the number of roles and symbols the same would make it difficult to interpret results presented in Tables 2 and 3, as we would no longer be able to differentiate between filler and roles properly. Having a smaller number of roles than fillers corresponds to having less number of grammatical roles than semantic concepts in language. We also ran experiments with different values of \lambda (regularization term) and observed that values higher than 10e-6 will decrease the final accuracy. We thus chose \lambda value to be a small value lower than this threshold. It still encourages R matrix to be orthogonal but not to the extent that it hurts performance. We also updated the regularization term to account for both over-complete and under-complete matrices in the new revision of the paper.
>
>
> We hope that the above explanations have addressed your concerns. We would be happy to provide more information regarding the experimental setup or results, should you have more questions.

---

### Public Comment · ~Florian_Mai1 · 2019-10-01
**An alternative interpretation of HUBERT's results**

Let me try to give alternative answers to some of the questions you pose in the beginning of Section 4.

Question: "Does adding a TPR layer (as in the previous section) on top of BERT impact its performance positively or negatively?"

Your answer:
The TPR layer positively affects the results, because the dev-set performance on MNLI improves by 0.7 over plain BERT.

Alternative answer:
There are many issues with your analysis: First, 0.7 points of improvement can easily be explained with BERT's susceptibility to random seeds. As far as I see, you didn't control for that. Second, you claim that just adding an LSTM on top of BERT is not enough, only TPRs can do the trick. But putting an LSTM on top already accounts for 0.45 points of the 0.72 improvement. How can you be sure that the additional improvement actually come from TPRs, and not just from, say, more parameters? Third, your choice of MNLI for answering this question was very selective. In fact, the results of "Baseline Acc" in Table 2 show essentially the results of what your report in Table 1, but on other datasets. Here, the performance of HUBERT drops considerably compared to BERT.

Question: Does transferring role (R) and/or symbol (S) embeddings (described in the previous section) improve transfer learning on BERT across the GLUE tasks?

Your answer: Yes, because the "Gain" column in Table 2 and 3 has more and larger positive values for HUBERT than for BERT.

Alternative answer:
No, since the values of the "Gain" column are only larger because the "baseline" results are much worse than is the case in BERT. In absolute numbers, i.e., the "Finetuned-Accuracy" column, the performance of HUBERT is comparable to BERT (sometimes better, sometimes worse). This is despite the fact that you effectively used 7 times as many random seeds for HUBERT than for BERT: You chose the results that were best after finetuning depending on what components you transfer.

Question: Is the ability to transfer the TPR layer limited to GLUE tasks?

Your answer: No, because you also observe better results on the HANS dataset.

Alternative answer:
Comparing only BERT and HUBERT, HUBERT does not seem to do better than BERT. The 62 percentage points improvement of HUBERT+ is unbelievably high, which must either be a mistake or can be explained by the fact that you pretrained on SNLI. Unfortunately, you don't have a variant where you pretrain BERT on SNLI, so we cannot know.

Question: Does transferring the BERT model parameters finetuned on one GLUE task help the other tasks in the Natural Language Understanding (NLU) benchmarks?

Your answer: No, because the "Gain" column doesn't have positive results.
Alternative answer: Your results suggest no, but you failed to acknowledge two important papers that have looked at the same question: BERT on STILTS, https://arxiv.org/pdf/1811.01088.pdf , and Can you tell me how to get past sesame street?, https://arxiv.org/pdf/1812.10860.pdf , both finding positive effects of choosing GLUE tasks (esp. MNLI) as intermediate tasks.


I think your model is interesting, and the goal to increase the linguistic intelligence of BERT is an important one, but your results do not at all match what you claim. I think your approach to measure linguistic intelligence is suboptimal in the first place: We are not so much interested in whether the peak performance on a close-to-solved benchmark improves after training on an intermediate task - we are more interested in whether your model gets to better results more quickly, i.e., with fewer training examples. To this end, you should rather consider how Yogatama et al. define linguistic intelligence in this paper: https://arxiv.org/pdf/1901.11373.pdf . If your model can improve on this metric, it will be an important result.

---

> ### Author Response · Authors · 2019-10-07
> **Response to reader comments**
>
> Thank you Florian for reading our paper and for your suggestions.
>
> I will respond to each of the points you raised below: (I will only include the questions and my response because of space limitation on Openreview)
>
> Question: "Does adding a TPR layer (as in the previous section) on top of BERT impact its performance positively or negatively?"
>
> Response:
> First point: In fact for the results in Table 1, we trained each model using 3 different seeds and also performed hyper-parameter tuning. We then chose the model with the best accuracy on the dev set. We, however, did not perform hyper-parameter tuning for transfer learning experiments due to time and computation resources limit.
>
> Second point: In our initial experiments we performed an ablation study by inserting a TPR or LSTM layer on top of certain layers of BERT (e.g. first 2 layers) and omitting the remaining layers. In those experiments, we observed that LSTM was degrading the performance whereas TPR was improving it. However, this conclusion is not True when all 12 layers of bert-base are used in this experiment. We will revise that section again and omit "adding an LSTM on top of BERT is not enough".
>
> Third point: Although this is true, as mentioned in the paper, the point of this experiment was not to claim better results on a specific task. It mostly serves as a sanity-check for HUBERT. We did experiment with another corpus (QQP) in our transfer learning experiments to control for that effect. We are planning to run more experiments using other tasks as a source in the future.
>
>
> Question: Does transferring role (R) and/or symbol (S) embeddings (described in the previous section) improve transfer learning on BERT across the GLUE tasks?
>
> Response:
> Although the baseline results for HUBERT are slightly worse than BERT in Table 2, we show that HUBERT performs much better when initialized with a fine-tuned model (compared to its own baseline), whereas BERT sometimes degrades the performance after knowledge transfer.
> We controlled for the randomness by fine-tuning BERT with 3 different seeds and choosing the best results among them. However, for HUBERT we used the same seed for all 7 experiments and only changed the initializations (as described in the paper).
>
> Question: Is the ability to transfer the TPR layer limited to GLUE tasks?
>
> Response:
> We do now! We ran more experiments testing BERT and HUBERT on HANS after paper submission. Our baseline BERT results were actually higher than what was reported in the paper (which might be due to different hyper-parameters.) We are still observing huge improvements in lexical and considerable improvements in constituent heuristics. We only see a small drop (1%) for subsequence heuristic. The results for BERT are alarming though. After fine-tuning (the model which is pre-trained on MNLI) on SNLI the accuracy drops 2%, 6%, and 16% for lexical, subsequence, and constituent heuristics, respectively. We will update our paper based on the current results.
>
>
> Question: Does transferring the BERT model parameters finetuned on one GLUE task help the other tasks in the Natural Language Understanding (NLU) benchmarks?
>
> Response:
> Thanks for suggesting those papers. We will make sure to cite those in the next revision of the paper.
> Although there are some positive trends in the results, after taking a careful look, we found that the results do not follow a consistent pattern when using different corpora for fine-tuning BERT, and often degrades downstream transfer. Even for data-rich tasks like QNLI, regardless of the intermediate task and multi-tasking strategy, the baseline results do not improve. In fact, in "Can you tell me how to get past sesame street?" paper, the authors mention in the abstract that: "our results are mixed across pretraining tasks and show some concerning trends: ... In addition, fine-tuning BERT on an intermediate task often negatively impacts downstream transfer." HUBERT, on the other hand, consistently shows improvements over the baseline after fine-tuned on intermediate tasks like MNLI and QQP. It achieves that goal by removing data-specific semantics from BERT representations and only transferring sentence structure (grammar) which is shared among tasks.
>
>
> I think your model is interesting, ...
>
> Response:
> Our goal in this work is understanding and improving transfer learning in Natural Language tasks. We suggest and show that current SOTA models are not able to efficiently transfer knowledge learned from one task/ domain to other tasks/ domains. To alleviate this problem we propose untangling semantics and structure of the learned representations by the means of TPR and only transferring those instead of the whole model.
> We have multiple ideas for follow-up works. What you suggested is also an interesting topic which we would like to explore more.
>
> We thank you again for spending the time reading our paper and for your insightful suggestions and comments.
>
> Sincerely,
> Authors

---

### Author Response · Authors · 2019-11-15
**Response to all the reviewers + Summary of the revisions**

We want to thank all the reviewers for their constructive feedback and helpful comments.

One major concern addressed by all reviewers is the interpretability of global Role and Filler matrices.
To address this, we collected the POS tags and the attention vectors over the R matrix (embeddings of individual roles) for each token in the sentence. The attention scores are the distribution over the possible roles for a specific token. For each token, we chose two roles that have the highest attention scores and represent them as a tuple. We then found the distribution of the roles chosen for a specific POS tag. Our preliminary results show that there are correlations between some of the roles and POS tags showing that the learned roles can be indicative of grammar structures.

We have revised our paper and have submitted the new manuscript for your review. We have highlighted the parts that have been changed for better reading. Below is a summary of changes made in the new revision:

1) We have improved the readability of our paper and made our claims in Section 4 more clear.
2) We have devoted a part in Section 2 to explain the previous work on BERT and fine-tuning methods such as STILTs (https://arxiv.org/abs/1811.01088).
3) We have done major revision on Section 4.3 by adding more results and explanations, to make the comparison between BERT+ and HUBERT+ fairer. HUBERT+ (HUBERT fine-tuned on MNLI and then SNLI subsequently) is outperforming BERT on all challenging non-entailment cases setting a new state-of-the-art average accuracy of 63.22% which is 2.21% higher than the BERT's average accuracy.
4) We have added more explanations on the experiment setting and implantation details in Section A.2. This should address the concerns regarding the variance in the results by reviewer 1 and 2.
5) We have added a new section in Appendix (A.4) on the interpretation of the learned roles.

---

### Decision · Program_Chairs · 2019-12-19

**Decision:**

Reject

**Comment:**

The paper introduces additional layers on top BERT type models for disentangling of semantic and positional information.  The paper demonstrates (small) performance gains in transfer learning compared to pure BERT baseline.

Both reviewers and authors have engaged in a constructive discussion of the merits of the proposed method. Although the reviewers appreciate the ideas and parts of the paper the consensus among the reviewers is that the evaluation of the method is not clearcut enough to warrant publication.

Rejection is therefore recommended. Given the good ideas presented in the paper and the promising results the authors are encouraged to take the feedback into account and submit to the next ML conference.